# Linking Distributed Optimization Models for Food, Water, and Energy Security Nexus Management

Yuri Ermoliev [1], Anatolij G. Zagorodny [2], Vjacheslav L. Bogdanov [3], Tatiana Ermolieva [1,*], Petr Havlik [1], Elena Rovenskaya [1], Nadejda Komendantova [1] and Michael Obersteiner [1]

1 International Institute for Applied Systems Analysis, 2361 Laxenburg, Austria; ermoliev@iiasa.ac.at (Y.E.); havlik.petr@gmail.com (P.H.); rovenska@iiasa.ac.at (E.R.); komendan@iiasa.ac.at (N.K.); oberstei@iiasa.ac.at (M.O.)
2 Bogolyubov Institute for Theoretical Physics, National Academy of Sciences of Ukraine, 03142 Kiev, Ukraine; Zagorodny@nas.gov.ua
3 Timoshenko Institute of Mechanics, National Academy of Sciences of Ukraine, 03142 Kiev, Ukraine; Bogdanov@nas.gov.ua
* Correspondence: ermol@iiasa.ac.at; Tel.: +43-2236-807581

**Abstract:** Traditional integrated modeling (IM) is based on developing and aggregating all relevant (sub)models and data into a single integrated linear programming (LP) model. Unfortunately, this approach is not applicable for IM under asymmetric information (ASI), i.e., when "private" information regarding sectoral/regional models is not available, or it cannot be shared by modeling teams (sectoral agencies). The lack of common information about LP submodels makes LP methods inapplicable for integrated LP modeling. The aim of this paper is to develop a new approach to link and optimize distributed sectoral/regional optimization models, providing a means of decentralized cross-sectoral coordination in the situation of ASI. Thus, the linkage methodology enables the investigation of policies in interdependent systems in a "decentralized" fashion. For linkage, the sectoral/regional models do not need recoding or reprogramming. They also do not require additional data harmonization tasks. Instead, they solve their LP submodels independently and in parallel by a specific iterative subgradient algorithm for nonsmooth optimization. The submodels continue to be the same separate LP models. A social planner (regulatory agency) only needs to adjust the joint resource constraints to simple subgradient changes calculated by the algorithm. The approach enables more stable and resilient systems' performance and resource allocation as compared to the independent policies designed by separate models without accounting for interdependencies. The paper illustrates the application of the methodology to link detailed energy and agricultural production planning models under joint constraints on water and land use.

**Keywords:** asymmetric information; linkage; nonsmooth optimization; subgradient; integrated modeling; food-energy-water-land nexus; machine learning

## 1. Introduction

The increasing interdependencies among food-energy-water-environmental (FEWE) sectors require integrated coherent planning and coordinated policies for sustainable development and security nexus. The sectors become more interconnected because they utilize common, often rather limited, resources, both natural (e.g., land, water, air quality) and socio-economic (e.g., investments, labor force). For example, land and water are needed not only for agricultural production but also for hydropower generation, coal mining and processing, power plants cooling, renewable energy, and hydrogen production.

The energy sector is one of the largest and fast-growing water consumers. The more water is used by the energy sector, the more vulnerable energy production and production in other water-dependent sectors, becomes [1]. Climate change concerns and rapid energy sector transition towards renewable energy sources tighten the links between energy and

agricultural markets. Agricultural commodities have become an important energy resource because of biofuels mandates. Vulnerability of crop yields, increasing grain demand, and price volatility directly and indirectly influence the market for fuel transportation and transportation costs [2]. At the same time, crude oil, gas, and electricity markets and prices have an effect on agricultural production costs and prices [2–4].

Additional linkages and interactions in FEWE systems emerge due to the introduction of new technologies, e.g., intermittent renewables, advanced irrigation, hydrogen production, water desalination, etc. The interdependencies can trigger systemic failures if sectoral policies ignore cross-sectoral interconnectedness [5]. The FEWE security nexus management requires an integrated approach to understand and deal with the numerous interactions between the FEWE systems [6]. This approach, compared to independent analysis, contributes immensely to sustainable development within and across sectors and scales.

Comprehensive sectoral models are being developed for planning and policy assessment in respective sectors. These models account for multiple details of sectoral production planning and resource utilization, including the analysis of factors and drivers determining demand, supply, and commodity price relationships [7,8]. For example, energy sector models investigate the interactions between renewables and fossil fuels, address energy market volatilities, and analyze the effects of new technologies and policy interventions to develop energy scenarios [7–12]. Land use and agricultural models support decision making regarding agricultural policies, analyze land potentials for the production of sufficient agricultural commodities to fulfill food security and biofuels mandates [13–15], and assess the effects of policy responses, including export bans and high export taxes, to cope with production shortfalls and offset increasing prices [2]. As a rule, these models consider and optimize sectoral goals accounting for respective production, demand, resource availability, and environmental quality constraints. Goals, production targets, resource demand, and quality in other sectors are hardly, if at all, accounted for. Thus, the limitation of the detailed sectoral models for FEWE security nexus lies in their restricted ability to consider dependencies and interactions beyond the defined sectoral system [2,3,12], e.g., cross-sectoral resource competition and joint production and demand relationships. Sectoral models cannot properly account for the objectives of a larger system. The feedbacks and interactions among FEWE systems are often analyzed through CGE (Computational General Equilibrium) and/or IAMs (Integrated Assessment models) [6]. These models, unfortunately, suffer from the lack of necessary details of sectoral models. They involve considerable simplifications and aggregations and, therefore, may not be sufficiently fit to provide insightful conclusions [16–19].

In this situation, the analysis of systemic regulations for FEWE security nexus can rely on distributed models' optimization and linkage methods enabling the establishment of relationships and dialogues between separate models of FEWE systems for the analysis of coordinated solutions without requiring to share or reveal systems-specific information.

In this paper, we consider the problem of linking sectoral and/or regional linear programming (LP) models into a cross-sectoral integrated model (IM) in the presence of joint constraints when "private" information regarding sectoral/regional models' is not available or cannot be shared by modeling teams (sectoral agencies), i.e., under asymmetric information (ASI). Such linkage is necessary for producing truly integrative management scenarios, especially when sectors utilize and compete for common resources or act under joint regulatory constraints or environmental mandates. The approach provides a means of decentralized cross-sectoral coordination and enables the investigation of policies in interdependent systems in a "decentralized" fashion. This facilitates more stable and resilient systems' performance and resource allocation as compared to the independent policies designed by separate models without accounting for interdependencies.

Cross-sectoral policy analysis in the presence of joint constraints can be addressed, e.g., with the generalized Nash equilibrium (GNE) approach [20]. Böhringer and Rutherford (2009) [21] consider linking energy system mathematical programming models into

a general equilibrium (GE) model of the overall economy. Ermoliev and von Winterfeldt (2012) [22] discuss the game-theoretic approaches, e.g., the Stackelberg leadership model, and their complexity due to the assumptions that each player (sector/region) has information about other players' goals and constraints. Traditional integrated deterministic optimization modeling also assumes full knowledge about the systems. It incorporates goals, individual and joint constraints, and data of all systems into a single code (hard integration), which can be considered as a multi-criteria optimization problem [23].

Our approach for linking separate optimization models under ASI is based on the parallel solving of equivalent nonsmooth optimization models by a simple iterative stochastic quasigradient (SQG) procedure [24,25] based on subgradients or generalised gradients [25–27] converging to an optimal welfare-maximizing linkage solution, i.e., to the solution of a "hard-integrated" model. This approach does not require sharing details about models' specifications. We can assume there is a network of distributed computers connecting computer models of a "social planner" (decision-makers or regulatory agencies), who attempt to achieve the best result for all sectors/regions (parties) involved. The linkage procedure can be interpreted as a kind of a "decentralized market system" [28]. According to this procedure, sectors/regions independently and in parallel optimize their goal functions under individual constraints without considering joint constraints. In general, joint constraints impose restrictions on total production, resource use, and emissions by all sectors/regions. The constraints can establish supply–demand relationships between the systems enabling the estimation of optimal production, resource use, and emission quotas for each system. The balance between the total energy (including biofuels) production and demand defines energy security; agricultural production and consumption reflect food security; total emissions and pollution constraints correspond to environmental security. The joint FEWE constraints satisfaction establishes the FEWE security nexus [29]. After independent optimization using initial approximations of various (e.g., production, resource use, emission) quotas, the sectors/regions provide the social planner with the information on their actual production, resource use, and respective shadow prices. The planner checks if the joint constraints are fulfilled. If not, i.e., there is "excess demand" or "excess supply" (i.e., total resource use, production, emissions by all systems are higher/lower than required), the planner revises the individual systems' quotas via shifting their current approximation in the direction defined by the corresponding dual variables. Thus, shadow prices signal systems to adjust their activities accordingly. Formally, the procedure is described in Section 2.3 and Appendix A.

In this way, the linkage allows us to avoid the "hard linking" of models in a single code, which is not possible because the systems do not want to share the information or because the individual models are too detailed and complex to be "hard-linked". The approach saves reprogramming efforts and allows parallel distributed (decentralized) computations of sectoral models instead of a large-scale integrated (centralized) model. This also preserves the original models in their initial state for other linkages. The use of detailed sectoral and regional models instead of their aggregated simplified versions also enables us to account for critically important local details. Similar computerized decentralized "negotiation" processes between distributed models (agents) have been developed for the design of robust carbon trading markets (e.g., [30] and references therein) and for the allocation of water quotas (e.g., [31]). The linkage procedure can be considered as a new machine learning algorithm, namely, as a general endogenous reinforced learning problem of how software agents (models) take decisions in order to maximize the cumulative reward (total welfare) [32].

The paper is organized as follows. Section 2 discusses the problem of models' linkage under joint constraints. Section 2.1 presents a short overview and the main shortfalls of several existing approaches, Section 2.2. formulates the problem of distributed LP models' linkage in the presence of joint resource constraints and ASI, and Section 2.3 outlines the linkage solution procedure based on the parallel solving of equivalent nonsmooth optimization model following a simple iterative subgradient algorithm. The details and

main properties of the algorithms are presented in the Appendix A. Section 3 illustrates the application of the methodology to link detailed energy and agricultural production planning models under joint constraints on water and land use. In addition, the joint constraints can impose restrictions on total energy production by the energy sector (electricity, gas, diesel, etc.) and land use sector (biodiesel, methanol); total energy use by energy and agricultural sectors; total agricultural production by distributed farmers/regions, etc. Section 4 concludes and outlines potential further extensions of the approach, for example, to include more details of energy and natural resources dynamics in general.

## 2. Linking Distributed Optimization Models under Joint Resource Constraints

### 2.1. Social Equilibrium Game Approach

In the absence of coordination between systems (sectors, regions), they can act selfishly and aim at maximizing their own objective function. They attempt to secure as high resource quotas as possible. Such a situation can be modeled using the non-cooperative game-theoretic framework. For example, social equilibrium games [20] have been formulated to include joint constraints. The generalized Nash equilibrium (GNE) solution, if it exists, allocates production and resources among systems (sectors/regions), fulfilling the joint constraint. However, the decisions are made independently, and collective efforts for managing common resources are ignored. Importantly, the existence, uniqueness, and stability of the GNE, as well as a realistic large-scale implementation of this concept, cannot be guaranteed, as emphasized by Harker (1991) [20]. Moreover, in [20], it is highlighted that the GNE solutions set are rarely connected. Hence, a complete analysis of equilibriums, in this case, is a complex task, requiring additional assumptions.

The analysis can become even more complex if the joint constraints are based on the equilibrium (optimality) conditions arising from the problem formulated in the form of a principal-agent game or a leader-follower Stackelberg game [22]. For example, in the case of nonsmooth goal functions required for linking systems under ASI (distributed models' optimization), the use of optimality conditions would require implicit sets of generalized gradients. Due to the computational complexity, heuristic methods are often used; however, they lack rigorous convergence proof.

Linking bottom-up mathematical programming models of the energy system into a top-down general equilibrium model of the overall economy is discussed by Böhringer and Rutherford in [21]. The paper shows that the formulation of market equilibrium conditions using complementarity equations permit the integration of models, but the convergence of the iterative procedure integrating the models cannot be guaranteed. In specific cases, models of general equilibrium are reduced to optimization problems [33].

Ermoliev and von Winterfeldt (2012) [3] demonstrate that the complexity of the game-theoretic approaches is due to quite unrealistic assumptions that each player (sector/region) is in possession of the knowledge on exact and unique responses of other players. Therefore, even in the simplest linear cases, this assumption leads to extremely complex discontinuous problems. More realistic assumptions of uncertain response functions in combination with a concept of robust decisions results in stable large-scale solutions.

There exists a vast literature on important problems and methods for distributed systems' optimization under joint constraints, e.g., optimal control and economic dispatch in smart grids [34], agricultural production planning for the multi-farmer systems [35], network optimization [36–38], and optimal transportation problems [39–41]. Yet, these approaches consider the optimization of a total objective function representing a sum of individual objective functions of the involved systems. Thus, the problems assume full information regarding the systems is available to a social planner. They are formulated similarly to traditional integrated "centralized" optimization modeling, combining goals, constraints, and data of all models into a single code.

Our problem is more complex as it deals with the coordination of decentralized systems' models in the presence of joint constraints and ASI. In this case, the approach is based on a specific iterative nonsmooth optimization procedure (see Sections 2.2 and 2.3

and Appendix A). As we noted, the integrated solution of separate LP models under ASI cannot be accomplished by LP methods. In Section 2.2, we formulate the problem of distributed systems optimization in the presence of joint resource constraints under ASI, and in Section 2.3, we present the models' linkage algorithm.

### 2.2. LP Models under Joint Constraints

Let us formulate the basic problem of separate sectoral or regional LP models optimization under joint resource constraints. Consider separate models of $K$ systems in the following LP form:

$$\left\langle c^{(k)}, x^{(k)} \right\rangle \rightarrow max \tag{1}$$

$$x^{(k)} \geq 0 \tag{2}$$

$$A^{(k)} x^{(k)} \leq b^{(k)} \tag{3}$$

where components of vector $x^{(k)}$ are variables to be determined, vector $b^{(k)}$ defines system-specific demand or resource constraints, and vector $c^{(k)}$ corresponds to net unit profits, $k = 1, 2, \ldots, K$. The dependence of system $k$ on common resources are defined by constraint (4)

$$B^{(k)} x^{(k)} \leq y^{(k)} \tag{4}$$

where $y^{(k)}$ defines resource quota allocated to system $k$. Therefore, Formula (3) represents system-specific constraints and Formula (4) establishes systemic relations among systems by allocating quotas $y^{(k)}$. The quotas $y^{(k)}$ fulfil the joint resource constraint on the use of common resources

$$\sum_{k=1}^{K} D^{(k)} y^{(k)} \leq d, \tag{5}$$

where matrix $D^{(k)}$ defines the marginal resource use by system $k$ and $d$ is the total available resource, $d \geq 0$. Thus, each system $k$ maximizes its objective function (1) by choosing $x^{(k)}$ and $y^{(k)}$ from the feasible set defined by (2) and (3), so that (4) and (5) are also fulfilled.

In the presence of full information regarding a system, the problem of models' linkage can be formulated and solved by a central planner (regulator) as a total net profit maximization

$$\sum_{k=1}^{K} \left\langle c^{(k)}, x^{(k)} \right\rangle \rightarrow max \tag{6}$$

s.t. to constraints (2)–(5), $k = 1, 2, \ldots, K$. In this model, the net profits are defined as the amount of money left after subtracting production costs from the total profit. In a more general case, the net profits can account for taxes, interest, and other expenses.

However, when the information on $b^{(k)}, c^{(k)}, A^{(k)}, B^{(k)}, x^{(k)}$ of system $k$ is not available to the planner, the integrated LP model (2)–(6) under ASI cannot be solved by LP method due to the lack of common information about submodels.

We propose the consistent approach for linking distributed optimization models under ASI based on the parallel solving of equivalent nonsmooth optimization models following a simple iterative subgradient algorithm. The convergence and other properties of the algorithm are presented in Appendix A. The proposed linkage approach does not require full, common information regarding the models' specification, and it can be seen as an endogenous reinforced learning algorithm describing how distributed agents (models) can make decisions to maximize the "cumulative reward". Section 2.3. outlines the algorithm.

### 2.3. Nonsmooth Model and Linking Algorithm

The basic nonsmooth optimization model under ASI can be solved by a specific iterative subgradient linkage algorithm. For a given vector $y = \left( y^{(1)}, \ldots, y^{(K)} \right)$ let us denote by $F(y)$ the optimal value of function (6) under constraints (2)–(4). Therefore,

$F(y) = \sum_{k=1}^{K} f^{(k)}(y)$, where $f^{(k)}(y) = \left( c^{(k)}, \left( x^{(k)}(y) \right) \right)$ are concave nonsmooth functions. In this function $x^{(k)}(y)$ are optimal solutions of (1)–(4).

The required linkage algorithm is defined as a subgradient procedure maximizing function $F(y)$ s.t. the joint constraints (5). These constraints identify the feasible set of the algorithm, which can be denoted as $Y$. Therefore, an optimal solution maximizing $F(y)$, $y \in Y$, also defines an optimal linkage or a solution of the integrated LP model under ASI. In the following, we assume the existence of solutions $x^{(k)}(y)$, $y \in Y$, for all $k$.

The linkage algorithm can be summarized as follows. Imagine there is a network of distributed computers connecting submodels, say sectors, with a computer of a social planner. At the initial step, sectors $k$, $k = 1, \ldots, K$, use arbitrary chosen vectors $y^{0(k)}$ of resource quotas. They submit the information on $y^{0(k)}$ to the central computer. The computer updates quotas $y^0 = \left( y^{0(1)}, \ldots, y^{0(K)} \right)$ by projecting them onto set $Y$, defining a first feasible approximation $y^1 = \left( y^{1(1)}, \ldots, y^{1(K)} \right)$. All sectors independently solve models (1)–(4) with resource quotas $y^1$, calculate shadow prices $v^{1(k)}$ of common resources (constraint (4)), and submit them to the central computer. The central computer calculates $y^1 + \rho_1 v^1$ with a step-size $\rho_1$ such that the product $\rho_1 v^1$ corresponds to the scale of $y^1$. Vector $y^1 + \rho_1 v^1$ is projected onto $Y$ to derive quotas $y^2$. At the iteration $s+1$, the algorithm derives the next approximation of quotas $y^{s+1} = \left( y^{s+1(1)}, \ldots, y^{s+1(K)} \right)$ by shifting $y^s$ in the direction of vector $v^s = \left( v^{s(1)}, \ldots, v^{s(K)} \right)$, according to the following procedure.

$$y^{s+1} = \pi_Y(y^s + \rho_s v^s), \ s = 1, 2, \ldots, \tag{7}$$

where $\rho_s$ is a step-dependent multiplier, which is a method's parameter, $\pi_Y(\cdot)$ is the orthogonal projection operator onto set $Y$ defined by (5). Vector $v^s$ is a generalized gradient or a subgradient of function $F(y)$ at $y = y^s$. The step-size $\rho_s$ is chosen from rather general and natural requirements: $\rho_s \geq 0$, $\rho_s \to 0$, $\sum_{s=1}^{\infty} \rho_s = \infty$, (e.g., $\rho_s = 1/s$), because subgradients (generalized gradients) are not, in general, the increasing directions of functions.

At each iteration, all sectors independently calculate stopping criteria $\varepsilon_k(s) = \left( b^{(k)}, u^{s(k)}(y^s) \right) + \left( y^{s(k)}, v^{s(k)}(y^s) \right) - w_k \left( c^{(k)}, x^{s(k)}(y^s) \right)$ and submit values $\varepsilon_k(s)$ to the central computer. If $\sum_k \varepsilon_k(s) \leq \varepsilon \geq 0$, where $\varepsilon$ is an admissible accuracy, then the algorithm stops. Otherwise, it continues further.

The convergence theorem shows that the parallel independent optimization (linkage) of sectoral/regional models according to this algorithm without revealing sectoral/regional information is possible due to the additional requirement $\sum_s \rho_s^2 < \infty$. This allows us to prove the convergence of solutions (linkages) $y^s$ rather than the convergence of objective function $F(y^s)$. The convergence of the proposed linkage algorithm under ASI is based on the theory of (continuously) non-differentiable optimization. The details of convergence theorem, stopping criterion, subgradients, and computing projections can be found in Appendix A.

### 3. Linking Energy and Agricultural Models for Food-Energy-Water Nexus

The proposed iterative algorithm has been applied for linking energy and agricultural sectoral models under joint constraints on water and land use. Both models can be used for optimal energy and agricultural production and allocation planning. In the following, we only briefly outline the models. Further details can be found, for example, in [9–11,13–15]. The models are spatially explicit, which allows us to link the models across locations and thus control local drivers having significant implications on the overall results of models' integration.

The energy model incorporates the main stages of energy flows from resources to demands: energy extraction from energy resources, primary energy conversion into secondary energy forms, transport and distribution of energy to the point of end, and conversion into products for end-users to fulfill specific demands. The structure of the model is such that it

can incorporate various energy resources, e.g., coal, gas, crude oil, renewables. Primary energy sources include coal, crude oil, gas, solar, wind, etc.; secondary energy sources are fuel oil, methanol, hydrogen, electricity, ammonia, etc.; final energy products are coal, fuel oil, gas, hydrogen, ammonia, methanol, electricity, etc. Demands for useful energy products come from main sectors of the economy: industrial, residential, transport, agricultural, water, and energy. Each technology is characterized by uniting costs, efficiency, lifetime, emissions, etc. Additional sectoral (and cross-sectoral joint) constraints are imposed to capture the requirements and the limitations on natural resource use and availability and investments. The model can include existing technologies, as well as new zero-carbon green technologies, at the beginning of implementation or even in the research stage, e.g., various renewable and carbon-capturing technologies.

The agricultural model includes main crops and livestock production and management systems, characterized by systems-specific production costs, water and fertilizer requirements, emission factors, and other parameters. The supply of crops and livestock products need to cover food, feed, and biofuel demands and fulfill security constraints. The food security constraint requires that the energy and nutrients consumption from grain and livestock products is not less than the required kilocalories and nutrients needed to satisfy dietary requirements in cereals, vegetables, and animal products (meat and dairy products). Livestock feeds fulfill livestock dietary requirements in energy intake measured in megacalories. Biofuel production from crops (and agricultural residues) must fulfill biofuel mandates. In the model, land uses comprise agricultural (crop and pasture) land, grassland, and natural land. Land use changes can be regulated by setting regulatory constraints on land expansion and conversion. Security constraints introduce competition for limited natural resources (land and water) among different land uses.

Energy and agricultural sectors compete for common land and water resources. Assume that regional planners, decision makers, and sectoral authorities pursue a goal to minimize costs and maximize profits from energy and agricultural production under various joint balance (supply-demand) and resource constraints to fulfill the energy and agricultural demands. Namely, the goal is to choose a portfolio of energy technologies to be installed and operated to produce, convert, and transfer energy products among locations; and a portfolio of agricultural technologies and management systems to produce and transfer among locations agricultural commodities fulfilling constraints on natural resources, environmental pollution, and end-product demands. The models include relevant risk-related systems' performance criteria. These performance measures enable a better understanding of how systems (individually and jointly) might perform in the uncertain environment, in the presence of climate change, weather variability, market uncertainties, etc. A better understanding of how interdependent energy-water-agricultural systems may operate and how dangerous impacts of inappropriate decisions may be can motivate regional and sectoral planners, experts, and involved stakeholders in making cross-sectoral coherent and risk-adjusted robust decisions [9–11,14,15,22,29].

### 3.1. Energy, Water, and Agricultural Security Nexus in Shanxi Province, China

In the case study in Shanxi province, China [15], the energy model includes coal-based industries and processes, i.e., mining, washing, chemical production, and power generation. Most of the electricity in China comes from coal, which accounted for approximately 65% of the electricity generation mix in 2019. The coal-based technologies consume a vast amount of water, for example, for coal mining and washing, coal power plants cooling and steam production. About 51% of China's coal reserves lie in areas of high or extreme water scarcity, and about 30% are in water-stressed regions. Shanxi province is one of them.

The integrated energy-agricultural-water model is formulated as follows. A regional planner decides on the amount of coal $x_{ijlmt}$ and type $i$, $i = 1, \ldots, I$, to be extracted in location $j$, $j = 1, \ldots, J$, transported to location $m$, $m = 1, \ldots, M$, and converted by technology $t$, $t = 1, \ldots, T$. In addition, decisions $z_{kjm}$ are made concerning the amount of agricultural commodities $k$, $k = 1, \ldots, K$, to be produced in location $j$ and exported to

location $m$. The overall goal of the planner is to minimize the total costs related to energy (coal) and agricultural production, transportation, and conversion.

Individual sectoral goal functions are formulated as follows

$$\sum_{i,j,k,m,t} \left[ c_{ij}^{CP} + c_{ijm}^{CT} + c_{ijt}^{CC} \right] x_{ijmt} \rightarrow min \tag{8}$$

and

$$\sum_{i,j,k,m,t} \left[ c_{kj}^{AP} + c_{kj}^{AT} \right] z_{kjm} \rightarrow min \tag{9}$$

for energy (8) and agricultural (9) sectors. Production costs $c_{ij}^{CP}$ define all components of coal production costs, including extraction and washing, of a unit (tonne) coal of type $i$ in location $j$, transportation costs $c_{ijm}^{CT}$ represent all costs associated with transporting unit coal $i$ from location $j$ to location $m$, $c_{ijt}^{CC}$ define conversion costs of a unit coal $i$ by technology $t$ in location $j$, $c_{kj}^{AP}$ denote agricultural production cost per unit (tonne) agricultural commodity $k$ in location $j$, and $c_{kjm}^{AT}$ stands for the transportation costs of a unit agricultural commodity $k$ from $j$ to $m$, $i = 1, \ldots, I; j = 1, \ldots, J; m = 1, \ldots, M; t = 1, \ldots, T; k = 1, \ldots, K$. The energy model includes energy security constraints ensuring consumers demands for end products from coal, for example, electricity, heat, coke, gas, and oil:

$$\sum_{ijt} \alpha_{imt}^{d} x_{ijmt} \geq D_m^d \tag{10}$$

where $\alpha_{ijt}^{d}$ denotes the conversion efficiency of coal $i$ in location $j$ by technology $t$, the end product $d$, and $D_j^d$ stands for the end product $d$ demand.

Agricultural production is required to fulfil food security constraints defined by the sufficient kilocalories and nutrients provided to the population from agricultural commodities $k$ in location $m$:

$$\sum_{j} z_{kjm} \geq D_{km}^A \tag{11}$$

where $D_{km}^A$ is the required production of agricultural commodity $k$ in location $m$ to meet food security requirements, which can be calculated according to daily nutrients and calories requirements per capita approved by the World Health Organization (WHO).

Sectoral land use constraints

$$\sum_{i,m,t} x_{ijmt} \left( 1 - r_{ij} \right) \Delta l_j S_{ij} + g \sum_{i,m,t} x_{ijmt} \leq L_j^C \tag{12}$$

and

$$\sum_{k,m} l_{kj} z_{kjm} \leq L_j^A \tag{13}$$

incorporate land demand by coal (12) and crop (13) production activities, where $L_j^C$ and $L_j^A$ are land use constraints for coal and agricultural sectors in location $j$, respectively. In Equation (12), parameter $S_{ij}$ defines the area that can deteriorate (e.g., subside) as a result of coal mining of unit coal $i$ in location $j$, $\Delta l_j$ is a portion of agricultural land overlapping with a coal field in location $j$, land reclamation rate (or efficiency rate) is defined by parameter $r_{ij}$ for coal $i$ in location $j$, and $g_{ij}$ is a coal fraction that allows us to calculate the land under reject material. In Equation (13), parameter $l_{kj}$ defines the area required for a unit crop $k$ production in location $j$. Equation (14) introduces the restriction on the total land use in location $j$ by energy and agricultural sectors

$$\sum_{k,m} l_{kj} y_{kjm} + \sum_{i,m,t} x_{ijmt} \left( 1 - r_{ij} \right) \Delta l_j l_{ij} + g_{ij} \sum_{i,m,t} x_{ijmt} \leq L_j \tag{14}$$

Total available water $W_j$ for both sectors and sectoral water quotas ($W_j^E$ and $W_j^A$) significantly affect the choice of coal and crop (energy) production technologies through water utilization constraints:

$$\sum_{i,m,t} [w_{ij}^P + w_{ij}^d] x_{ijmt} \leq W_j^E \tag{15}$$

and

$$\sum_{k,m} w_{kj}^c z_{kmj} \leq W_j^A \tag{16}$$

where $W_j^E$ and $W_j^A$ are quotas on water use by coal and agricultural activities in location $j$, $w_{ij}^P$ defines water requirement for a unit coal $i$ production in location $j$, $w_{ij}^d$ is water required for a unit coal $i$ conversion in location $j$, and $w_{km}^c$ is water required for a unit crop $k$ production in location $j$. Water use $W_j^E$ for coal and $W_j^A$ for agricultural production are constrained by the total water $W_j$ available in $j$:

$$W_j^E + W_j^A \leq W_j \tag{17}$$

The models can be extended by including various other constraints, for example, water and air quality, $SO^2$ and $CO^2$ emissions targets, biofuel production mandates, etc.

In the condition of ASI, the planner does not have full information regarding separate LP energy ((8), (10), (12) and (15)) and agricultural ((9), (11), (13) and (14)) submodels. To link the models under joint constraints (14) and (17) we implement procedure (7). Thus, at the initial step s = 0, individual sectoral models are solved using initial sectoral land and water quotas $L_j^C(0)$, $L_j^A(0)$ and $W_j^C(0)$, $W_j^A(0)$. Resource quotas $y^s = (L_j^C(s), L_j^A(s), W_j^C(s), W_j^A(s))$ at step $s$ are adjusted according to (7) using shadow prices (dual variables) of energy and agricultural sectors land and water resource constraints

$$\sum_{i,m,t} x_{ijmt}(1 - r_{ij}) \Delta l_j S_{ij} + g \sum_{i,m,t} x_{ijmt} \leq L_j^C(s-1) \tag{18}$$

$$\sum_{k,m} l_{kj} y_{kjm} \leq L_j^A(s-1) \tag{19}$$

$$\sum_{i,m,t} w_{ij}^P x_{imlt} + \sum_{i,m,t} w_{ij}^d x_{ijmt} \leq W_j^C(s-1) \tag{20}$$

$$\sum_{k,m} w_{kj}^c y_{kmj} \leq W_j^A(s-1) \tag{21}$$

and constraints (14) and (17).

### 3.2. Selected Results

Results are compared for three cases: 1. Separately optimized energy and agricultural models; 2. The hard-linked integrated model (one-code model); 3. Separate models integrated via the linkage procedure (7). In case 1, the sectors are not restricted by joint resource constraints, and therefore the net profits can be higher than in cases 2 and 3; however, this is a misleading conclusion. In cases 2 and 3, results show that the iterative linkage process converges rather quickly. In 10 iterations, the optimal value of the integrated linked model (case 3) is almost equal to the optimal value of the integrated "hard-linked" model (case 2). In case 3, the proposed linkage algorithm allows to link models installed on remote computers through an iterative dialogue establishing optimal redistribution of water and land quotas among the sectors and locations.

Figure 1 illustrate the non-monotonic convergence of the linkage algorithm for three different scenarios of initial $y^0$ quotas allocated to energy and agricultural sectors. The

choice of the step-size $\rho_s$ in (7) affects the convergence rate, the value of the product $\rho_s v^s$ must correspond to the value of solutions $y^s$.

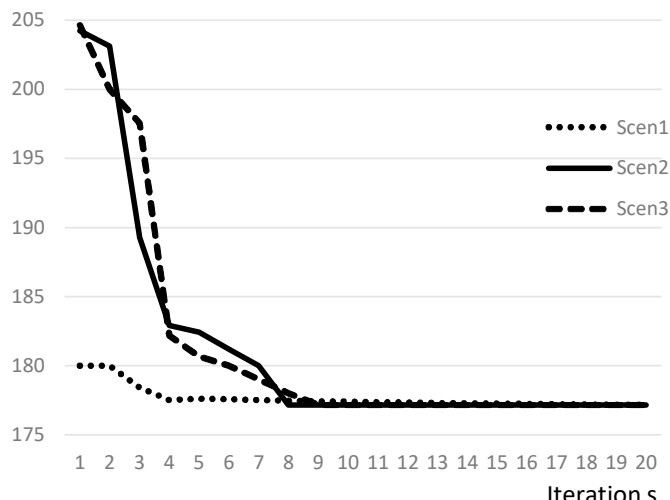

**Figure 1.** Convergence of the iterative linking procedure in terms of the goal function values $F(y^s)$. Vertical axis displays net profits; the iteration step is marked on the horizontal axis. The three curves (Scen1, Scen2, Scen3) correspond to three different initial land and water quota scenarios at s = 0.

## 4. Conclusions

In the paper, we consider the problem of linking separate distributed sectoral and/or regional optimization models into an inter-sectoral integrated model. The approach for linking models is based on an iterative algorithm that does not require models to exchange full information regarding their specifications. The resource quotas for each system and each resource are recalculated by systems independently and in parallel via shifting their current approximation in the direction defined by the corresponding dual variables from the primal sectoral optimization problem. In this way, the approach allows to avoid hard linking of the models in a single code that saves programming time and enables parallel distributed computations of sectoral models instead of a large-scale integrated model, i.e., addressing the well-known "curse of dimensionality" and large-scale data harmonization (management). This also preserves the original models in their initial state for other possible linkages. The proposed computational algorithm is based on an iterative stochastic quasigradient (SQG) procedure of, in general, nonsmooth nondifferentiable optimization converging to a socially optimal solution maximizing an implicit nested non-differentiable social welfare function. The convergence of the algorithm relies on the duality theory and non-differentiable optimization [20]. The iterative solution procedure can be used for robust estimation and machine learning problems. In particular, it can be viewed as an endogenous reinforced learning problem [32].

The iterative SQG-based methods and their stochastic versions are intended for the robust optimization of deterministic and stochastic systems with a large number of decision variables and scenarios of uncertainties due to the ability of these methods to link scenario-simulation and optimization procedures [42,43]. Therefore, the proposed method will be developed further for linking stochastic models enabling integrated management of global systemic risks, which are not detectable under traditional independent sectorial management ignoring cross-sectoral risk exposures. Fundamentally important to the possible extension of the presented method is the case of stochastic sectoral/regional models with interdependent systemic uncertainties and risks shaped by decisions of various agents. This includes the mitigation of floods by new land use decisions, for example, affecting flood scenarios. As a rule, this makes it impossible to separate scenario-generation and optimization processes calling for linking both simulation and optimization procedures in a way similar to algorithm (7), thus combining simulations of scenarios, new optimization

steps, new simulation of scenarios, and so on. In this case, we can determine new types of machine learning processes.

In the paper, we referred to linking regional and/or sectoral models. More generally, the problem can address the linkage of models at different spatial and temporal resolutions (e.g., local-global, considering more details of energy and natural resources dynamics in general). Therefore, the linkage problem can be formulated much more generally in terms of sub-models and integrated models, and the approach presented in this paper can still be applicable.

The linkage of models is, in a sense, opposite to decomposition methods. While in the decomposition (e.g., [44,45]), we split an existing integrated optimization model into a number of smaller sub-models, in the linkage, we obtain an integrated model of the system by linking existing explicitly unknown sub-models. The proposed linkage procedure provides flexibility, enabling the simultaneous use of linkage and decomposition procedures, in other words, endogenously disaggregating models to make their further integration (linkage) more efficient.

**Author Contributions:** Conceptualization, Y.E., A.G.Z., V.L.B., T.E., P.H., E.R., N.K. and M.O.; methodology, Y.E., T.E., E.R.; software, Y.E., T.E., E.R.; validation, Y.E., A.G.Z., V.L.B., T.E., P.H., E.R., N.K. and M.O.; formal analysis, Y.E., A.G.Z., V.L.B., T.E., P.H., E.R., N.K. and M.O.; investigation, Y.E., A.G.Z., V.L.B., T.E., P.H., E.R. and N.K.; data curation, T.E. and E.R.; writing—original draft preparation, Y.E., T.E., P.H., E.R. and M.O.; writing—review and editing, Y.E., A.G.Z., V.L.B., T.E., P.H., E.R., N.K. and M.O.; visualization, Y.E., T.E. and E.R. All authors have read and agreed to the published version of the manuscript.

**Funding:** This research was partially funded by EU projects COACCH (776479).

**Institutional Review Board Statement:** Not applicable.

**Informed Consent Statement:** Not applicable.

**Data Availability Statement:** The study does not report any data.

**Acknowledgments:** The development of linkage algorithms and case studies are part of methodological research of EU projects COACCH (776479) and a joint project between IIASA and National Academy of Sciences (Ukraine) on "Integrated robust management of food-energy-water-land use nexus for sustainable development".

**Conflicts of Interest:** The authors declare no conflict of interest.

## Appendix A
*Appendix A.1. Convergence*

Proposition (stopping criterion, subgradients): Assume there exist solutions $x^{(k)}(y)$ of all $K$ sectoral/regional models for feasible $y$ satisfying constraints (5). Then:

(a) Functions $f^{(k)}(y) = \left( c^{(k)}, x^{(k)}(y) \right)$, $F(y) = \sum_{k=1}^{K} f^{(k)} \left( x^{(k)} \right)$ are continuously concave non-differentiable functions for all $k$.

(b) The dual problem to (1)–(3) has a solution $\left( u^{(k)}(y), v^{(k)}(y) \right)$ for all $k$, and these solutions satisfy the stopping criterion of the linkage algorithm:

$$f^{(k)}(y) = \left( c^{(k)}, x(y) \right) = \left( b^{(k)}, u(y) \right) + (y, v(y)).$$

From this proposition follows the following important fact ([8,10,25]), which is fundamental for solving the linkage problem through maximizing non-differentiable function $F(y)$ using algorithm (7):

(c) For any feasible solution $z$ and $y$, $f^{(k)}(y) - f^{(k)}(z) \geq \left( v^{(k)}(y), y - z \right)$, that is, $v^{(k)}(y)$ is a subgradient of the concave non-differentiable function $f^{(k)}(y)$. Vector $v(y) =$

$\left(v^{(1)}(y), \ldots, v^{(K)}(y)\right)$ is a subgradient of function $F(y) = \sum_{k=1}^{K} f^{(k)}(y)$, $F_y(y) = v(y)$, that is, $F(y) - F(z) \geq (v(y), y - z)$.

Therefore, procedure (7) is a specific subgradient algorithm for maximizing the (continuously) non-differentiable concave function $F(y)$.

The following proposition shows that $y^s$ converges to an optimal linking vector $y^*$, maximizing $F(y)$ subject to joint constraints (5). Let us denote this feasible set by $Y$.

*Appendix A.2. Convergence Theorem (Non-Monotonic Convergence)*

Assume that
(1) The feasible set $Y$ is bounded;
(2) Step size $\rho_s$ satisfies the conditions:
$\rho_s \geq 0$, $\sum_{s=1}^{\infty} \rho_s = \infty$, $\sum_{s=1}^{\infty} \rho_s^2 < \infty$, say $p_s = 1/s$.
Then $\lim y^s \in Y^*$ for $s \to \infty$.
The following sequence of $\rho_s$ satisfies the conditions of the theorem: $\rho_s = \gamma_s/s$, $0 \leq \underline{\gamma} \leq \gamma_s \leq \overline{\gamma} < \infty$ for some positive constants $\underline{\gamma}$ and $\overline{\gamma}$.

*Appendix A.3. Computing the Projection*

The orthogonal projection $y^{s+1}$ of vector $y^s = y^s + \rho_s v^s$ onto $Y$ is calculated by minimizing the quadratic function $\| y^s + \rho_s v^s - y \|^2$ subject to joint constraints (5). This minimization is very fast due to $\rho_s v^s \to 0$, as vectors $v^s$ are bounded optimal dual solutions, and if $y^s$ is used as an initial approximation for $y^{s+1}$.

*Appendix A.4. Mixed Constraints*

Joint constraints (5) may have the following form:

$$\sum_{k=1}^{K} M^{(k)} x^{(k)} + \sum_{k=1}^{K} D^{(k)} y^{(k)} \leq \delta \tag{A1}$$

with some matrices $M^{(k)}$. Yet, problem (1)–(4) s.t. (A1) can be reformulated similar to problem (1)–(5). Let us define vectors $y^{(K+k)}$ such that $M^{(k)} x^{(k)} = y^{(K+k)}$, $k = 1, \ldots, K$. Now it is possible to rewrite (A1) as $\sum_{k=1}^{K} D^{(k)} y^{(k)} \leq \delta - \sum_{k=k+1}^{2K} y^{(k)}$ and after some renotation, derive the problem in the form (1)–(5).

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
