# Peer review of "Linking Distributed Optimization Models for Food, Water, and Energy Security Nexus Management"

_sustainability, doi:10.3390/su14031255_

Round 1

Reviewer 1 Report

The manuscript is on linking distributed optimization models for food, water, energy security nexus management. In this paper, the authors approach for linking models is based on an iterative algorithm that does not require models to exchange full information about their specifications. To reach their goal, they referred to linking regional and sectoral models.
The method of modeling is compatible with the problem and is affordable enough. The paper is well written and the process sounds correct. 

Author Response

Thank you for your high evaluation of our paper, we are very happy and pleased for the appreciation of our work!

Reviewer 2 Report

General comments:

  1. The approach to the investigated issues does not lead to market-related conclusions as expected initially from the title, i.e. optimization models for food, water and energy. In this sense, the overall intention of the authors fades away and although initially there is a good platform for investigation, results shown are relatively vague and do not resolve doubts described in the introductory section.  
  2. Given the rather complex issue investigated and the large number of interactions not only regarding methodology used but also affecting the evolution of the economic variables analysed, we miss a more critical and scientific position regarding energy and natural resources dynamics in general. This is a serious drawback in the document.
  3. Moreover, there is a clear need to provide enough background regarding the complex relationships among the three variables considered in this case, i.e. food , water and energy. . There is almost no reference to existing literature on this topic. Again this is serious drawback in the document.
  4. We miss a stepwise diagram or similar description explaining how the optimization process has been organized and judged.

Particular comments:

  1. Lines 39-42. Explaining the interaction dynamics of agricultural sectors and energy sectors is poor and misleading. One of the main factors behind the interaction, i.e. crude oil prices, is not well represented and assessed. We recommend to expand thoroughly on the interaction of agricultural and energy variables as it is at the heart of the investigation.  Some additional bibliography  to be analysed/included in the research would be:
    • Van Eyden, R.; Difeto, M.; Gupta, R.; Wohar, M.E. Oil price volatility and economic growth: Evidence from advanced economies using more than a century’s data. Appl. Energy 2019, 233–234, 612–621.
    • Foster, E.; Contestabile, M.; Blazquez, J.; Manzano, B. The unstudied barriers to widespread renewable energy deployment: Fossil fuel price responses. Energy Policy 2017, 103, 258–264.
    • Baffes, J.; Dennis, A. Long Term Drivers of Food Prices. In Working Paper 2013; World Bank: Washington, DC, USA, 2013.
    • Cansado-Bravo, P.; Rodríguez-Monroy, C. The Effects of Structural Breaks on Energy Resources in the Long Run. Evidence from the Last Two Oil Price Crashes before COVID-19. Designs 2020, 4, 49.
  2. Lines 52-55. It is stated that the on-going decentralization and deregulation processes ……… deciding their production plants ….. Can this reasoning be assessed and justified in view of existing research? Please support with evidence or delete
  3. Lines 90-91. It is stated that: the approach of our paper is the opposite. In a sense, we minimize the necessity to exchange information. There is not enough evidence supporting this claim. Please support with evidence or delete.
  4. Lines 97-99. Why is reasonable to assume this? Is there an example of a central hub computer of a central planner (regulator)?. Is it possible to know more about realizations of these arrangements building up the research later on?.
  5. Lines 145-154. It seems there is repetition of bibliography review and lines 78-84 in the Introduction. Please avoid repeating same topics and delete this second description.
  6. Lines 260-263. Please explain why the selected subsets add better value to the investigation than other categorizations. How coal can be either primary energy source and also secondary? How is the quota considered in a case like this?. How are solar and wind resources treated?, maybe through power generation capabilities?, please reorganize and explain solid foundations of this initial platform for subsequent research.
  7. Lines 285-287. Please add supporting evidence to the expression: … regional planners (…..) pursue a goal to minimize costs and maximize profits……Why should they do that. Please explain or delete.
  8. It is quoted along the document that ‘net profit’ is one main benchmarks for reference, however, there is no explanation or definition of this concept. This is again a very serious barrier to understand the document.

Author Response

Manuscript: sustainability-1524279

Title: Linking distributed optimization models for food, water, energy security nexus management

Authors: Yuri Ermoliev, Anatolij G. Zagorodny, Vjacheslav L. Bogdanov, Tatiana Ermolieva, Petr Havlik, Elena Rovenskaya, Nadejda Komendantova, Michael Obersteiner

Answers to Reviewer.

Thank you for the valuable comments and suggestions, which led to the improvement of our paper. Below please find detailed point-by-point explanations on how we addressed these comments and revised the paper.

General comments:

  1. The approach to the investigated issues does not lead to market-related conclusions as expected initially from the title, i.e. optimization models for food, water and energy. In this sense, the overall intention of the authors fades away and although initially there is a good platform for investigation, results shown are relatively vague and do not resolve doubts described in the introductory section.  

Answer:

Thank you for this valuable comment. We addressed this comment by expanding the discussion on the linkages and interdependencies between energy, water, agricultural variables, energy, and agricultural markets. For example, in the Introduction of the revised manuscript we added the following text: “The energy sector is one of the largest and the fast-growing water consumers [1]. The more water is used by energy sector, the more vulnerable become the energy production and the production in other water-dependent sectors [1]. Climate change concerns and rapid energy sector transition towards renewable energy sources tighten the links between energy and agricultural markets. Agricultural commodities become important energy resource because of biofuels mandates. Vulnerability of crop yields, increasing grains demand and price volatility influence, directly and indirectly, the market for transportation fuels and transportation costs [2]. At the same time, crude oil, gas and electricity markets and prices have effect on agricultural production costs and prices [2-3].

Additional linkages and interactions in FEWE systems emerge due to the introduction of new technologies, e.g., intermittent renewables, advanced irrigation, hydrogen production, water desalination, etc. The interdependencies can trigger systemic failures if sectoral policies ignore cross-sectoral interconnectedness [5]. The FEWE security nexus management requires integrated approach to understand and deal with the numerous interactions between the FEWE systems [6]. This approach, compared to independent analysis, contributes immensely to sustainable development within and across sectors and scales.

Comprehensive sectoral models are being developed for planning and policy assessment in respective sectors. These models account for multiple details of sectoral production planning and resource utilization including analysis of factors and drivers determining the demand, supply, and commodity price relationships [7-8]. For example, energy sector models investigate the interactions between renewables and fossil fuels, address energy market volatilities, analyze effects from new technologies and policy interventions, develop energy scenarios [7-12]. Land use and agricultural models support decision making regarding agricultural policies, analyze land potentials for production of sufficient agricultural commodities to fulfill food security and biofuels mandates [13-15], assess the effects of policy responses, including export bans and high export taxes, to cope with production shortfalls and offset increasing prices [2]. As a rule, these models consider and optimize sectoral goals accounting for respective production, demand, resource availability, and environmental quality constraints. Goals, production targets, resource demand and quality in other sectors are hardly, if at all, accounted for. Thus, the limitation of the detailed sectoral models for FEWE security nexus lies in their restricted ability to consider dependencies and interactions beyond the defined sectoral system [2, 3, 12], e.g., linkages between agricultural and energy markets, cross-sectoral resource competition, joint production and demand relationships. Sectoral models cannot properly account for the objectives of a larger system. The feedbacks and interactions among FEWE systems are often analyzed through CGE (Computational General Equilibrium) and/or IAMs (Integrated Assessment models) [6]. These models, unfortunately, suffer from the lack of necessary details of sectoral models. They involve considerable simplifications and aggregations and, therefore, may not be sufficiently fit to provide insightful conclusions [16-19].”

Also, we included the following discussion: “The linkage procedure can be interpreted as a kind of a "decentralized market system” [28]. According to this procedure, sectors/regions independently and in parallel optimize their goal functions under individual constraints, without considering joint constraints. In general, joint constraints impose restrictions on total production, resource use, and emissions by all sectors/regions. The constraints can establish supply-demand relationships between the systems enabling to estimate optimal production, resource use, or emission quotas for each system. The balance between the total energy (including biofuels) production and demand defines energy security; agricultural production and consumption reflect food security; total emissions and pollution constraints correspond to environmental security. The joint FEWE constraints satisfaction establishes the FEWE security nexus [29]. After the independent optimization using initial approximations of various (e.g., production, resource use, emission) quotas, the sectors/regions provide social planner with the information on their actual production, resource use, and respective shadow prices. The planner checks if the joint constraints are fulfilled. If not, i.e., there is “excess demand” or “excess supply” (i.e., total resource use, production, emissions by all systems are higher/lower than required), the planner revises the individual systems’ quotas via shifting their current approximation in the direction defined by the corresponding dual variables. Thus, shadow prices signal systems to adjust their activities accordingly. Formally, the procedure is described in section 2.3 and in Annex.”

  1. Given the rather complex issue investigated and the large number of interactions not only regarding methodology used but also affecting the evolution of the economic variables analysed, we miss a more critical and scientific position regarding energy and natural resources dynamics in general. This is a serious drawback in the document.

Answer: We very much value this comment, however, the paper does not aim to discuss the “energy and natural resources dynamics in general” which can be a topic for another paper. This paper develops a procedure to link and optimize distributed sectoral and/or regional optimization models thus providing a means of decentralized cross-sectoral coordination in the situation when “private” information about sectoral/regional models’ is not available or it cannot be shared by modeling teams (sectoral agencies). Thus, the linkage methodology enables to investigate policies in interdependent systems in a “decentralized” fashion. This will enable more stable and resilient systems’ performance and resource allocation as compared to the independent policies designed by separate models without accounting for interdependencies. 

The paper illustrates the application of the methodology to link detailed energy and agricultural production planning models under joint constraints on water and land use. In addition, the joint constraints can impose restriction on total energy production by energy sector (electricity, gas, diesel, etc.) and land use sector (biodiesel, methanol); total energy use by energy and agricultural sectors; total agricultural production by distributed farmers/regions, etc. Therefore, the discuss about “energy and natural resources dynamics in general” is beyond the scope of this paper and can be a topic of another paper.     

  1. Moreover, there is a clear need to provide enough background regarding the complex relationships among the three variables considered in this case, i.e. food , water and energy. . There is almost no reference to existing literature on this topic. Again this is serious drawback in the document.

Answer: We provide additional discussions regarding the links between agricultural, energy, water systems. Please, see the answer to Comment 1. In the revised version of the manuscript we include the suggested, and also additional, references on the interactions between the sectors. For example,

Carter, N. Energy’s water demand: Trends, vulnerabilities, and management. CRS (Congressional Research Service) Report for Congress, 7-5700, www.crs.gov, R41507, 2020 (available at https://digital.library.unt.edu/ark:/67531/metadc31387/).

Baffes, J., Dennis, A. Long Term Drivers of Food Prices. In Working Paper 2013; World Bank: Washington, DC, USA, 2013.

Taghizadeh-Hesary, F., Rasoulinezhad, E., Yoshino, N. Energy and food security: Linkages through price volatility. Energy Policy 2019, 128, 796-806.

Van Eyden, R., Difeto, M., Gupta, R., Wohar, M.E. Oil price volatility and economic growth: Evidence from advanced economies using more than a century’s data. Appl. Energy 2019, 233–234, 612–621.

Grafton, Q., McLindin, M., Hussey, K., Wyrwoll, P., Wichelns, D., Ringler, C., Garrick, D., Pittock, J., Wheeler, S., Orr, S., Matthews, N., Ansink, E., Aureli, A., Connell, D., De Stefano, L., Dowsley, K., Farolfi, S., Hall, J., Katic, P., Lankford, B., Leckie, H., McCartney, M., Pohlner, H., Ratna, N., Rubarenzya, M.-H., Sai Raman, S.-N., Wheeler, K., Williams, J. Responding to global challenges in Food, Energy, Environment and Water: Risks and options assessment for decision-making. Asia & the Pacific Policy Studies 2016,  3/2, 275–299, doi: 10.1002/app5.128

Howells, M., Hermann, S., Welsch, M., Bazilian, M., Segerstrom, R., Alfstad, T., Gielen, D., Rogner, H.-H., et al. Integrated analysis of climate change, land-use, energy and water strategies. Nature Climate Change 2013, 3 (7), 621-626. 10.1038/nclimate1789.

Foster, E., Contestabile, M., Blazquez, J., Manzano, B. The unstudied barriers to widespread renewable energy deployment: Fossil fuel price responses. Energy Policy 2017, 103, 258–264.

Cansado-Bravo, P., Rodríguez-Monroy, C. The Effects of structural breaks on energy resources in the long run. Evidence from the last two oil price crashes before COVID-19. Designs 2020, 4, 49.

Gambhir A., Butnar, I., Li, P-H., Smith, P., Strachan, N. A review of criticisms of Integrated Assessment Models and proposed approaches to address these through the lens of BECCS. Energies 2019.

Bosetti, V.; Marangoni, G.; Borgonovo, E.; Diaz Anadon, L.; Barron, R.; McJeon, H.C.; Politis, S.; Friley, P. Sensitivity to energy technology costs: A multi-model comparison analysis. Energy Policy 2015, 80, 244–263.

Gielen, D.J., Gerlagh T., Bos, A.J.M. MATTER 1.0 - A MARKAL Energy and Materials System - Model Characterisation. ECN report ECN-C-98-085; ECN, Petten, the Netherlands, 1998.

Doukas, H.; Nikas, A.; González-Eguino, M.; Arto, I.; Anger-Kraavi, A. From Integrated to integrative: Delivering on the Paris Agreement. Sustainability 2018, 10, 2299.

  1. We miss a stepwise diagram or similar description explaining how the optimization process has been organized and judged.

Answer: The description of the iterative optimization (linkage) procedure and the “stopping” condition are presented in lines 215-247, pp.4-5, of the initial manuscript and in subsection 2.3., lines 255-297, pp. 5-6 in the revised manuscript. More details on the convergence properties and convergence theorem, stopping criterion, subgradients, and computing projections are provided in Annex.

Particular comments:

  1. Lines 39-42. Explaining the interaction dynamics of agricultural sectors and energy sectors is poor and misleading. One of the main factors behind the interaction, i.e. crude oil prices, is not well represented and assessed. We recommend to expand thoroughly on the interaction of agricultural and energy variables as it is at the heart of the investigation. 

Answer: We expanded the discussion on the interactions between agricultural and energy sectors. In particular, “The energy sector is one of the largest and the fast-growing water consumers [1]. The more water is used by energy sector, the more vulnerable become the energy production and the production in other water-dependent sectors [1]. Climate change concerns and rapid energy sector transition towards renewable energy sources tighten the links between energy and agricultural markets. Agricultural commodities become important energy resource because of biofuels mandates. Vulnerability of crop yields, increasing grains demand and price volatility influence, directly and indirectly, the market for transportation fuels and transportation costs [2]. At the same time, crude oil, gas and electricity markets and prices have effect on agricultural production costs and prices [2-3].

Additional linkages and interactions in FEWE systems emerge due to the introduction of new technologies, e.g., intermittent renewables, advanced irrigation, hydrogen production, water desalination, etc. The interdependencies can trigger systemic failures if sectoral policies ignore cross-sectoral interconnectedness [5]. The FEWE security nexus management requires integrated approach to understand and deal with the numerous interactions between the FEWE systems [6]. This approach, compared to independent analysis, contributes immensely to sustainable development within and across sectors and scales.

Comprehensive sectoral models are being developed for planning and policy assessment in respective sectors. These models account for multiple details of sectoral production planning and resource utilization including analysis of factors and drivers determining the demand, supply, and commodity price relationships [7-8]. For example, energy sector models investigate the interactions between renewables and fossil fuels, address energy market volatilities, analyze effects from new technologies and policy interventions, develop energy scenarios [7-12]. Land use and agricultural models support decision making regarding agricultural policies, analyze land potentials for production of sufficient agricultural commodities to fulfill food security and biofuels mandates [13-15], assess the effects of policy responses, including export bans and high export taxes, to cope with production shortfalls and offset increasing prices [2]. As a rule, these models consider and optimize sectoral goals accounting for respective production, demand, resource availability, and environmental quality constraints. Goals, production targets, resource demand and quality in other sectors are hardly, if at all, accounted for. Thus, the limitation of the detailed sectoral models for FEWE security nexus lies in their restricted ability to consider dependencies and interactions beyond the defined sectoral system [2, 3, 12], e.g., linkages between agricultural and energy markets, cross-sectoral resource competition, joint production and demand relationships. Sectoral models cannot properly account for the objectives of a larger system. The feedbacks and interactions among FEWE systems are often analyzed through CGE (Computational General Equilibrium) and/or IAMs (Integrated Assessment models) [6]. These models, unfortunately, suffer from the lack of necessary details of sectoral models. They involve considerable simplifications and aggregations and, therefore, may not be sufficiently fit to provide insightful conclusions [16-19].”

  1. Some additional bibliography  to be analysed/included in the research would be:
    • Van Eyden, R.; Difeto, M.; Gupta, R.; Wohar, M.E. Oil price volatility and economic growth: Evidence from advanced economies using more than a century’s data. Appl. Energy 2019, 233–234, 612–621.
    • Foster, E.; Contestabile, M.; Blazquez, J.; Manzano, B. The unstudied barriers to widespread renewable energy deployment: Fossil fuel price responses. Energy Policy 2017, 103, 258–264.
    • Baffes, J.; Dennis, A. Long Term Drivers of Food Prices. In Working Paper 2013; World Bank: Washington, DC, USA, 2013.
    • Cansado-Bravo, P.; Rodríguez-Monroy, C. The Effects of Structural Breaks on Energy Resources in the Long Run. Evidence from the Last Two Oil Price Crashes before COVID-19. Designs 2020, 4, 49.

Answer: We included the suggested references in the discussion and in the list of references.

  1. Lines 52-55. It is stated that the on-going decentralization and deregulation processes ……… deciding their production plants ….. Can this reasoning be assessed and justified in view of existing research? Please support with evidence or delete

Answer: The paragraph has been deleted as it diverts the overall discussion focused on the linkage procedure.

  1. Lines 90-91. It is stated that: the approach of our paper is the opposite. In a sense, we minimize the necessity to exchange information. There is not enough evidence supporting this claim. Please support with evidence or delete.

Answer: The sentence has been deleted as it does not fit well into the discussion.

  1. Lines 97-99. Why is reasonable to assume this? Is there an example of a central hub computer of a central planner (regulator)?. Is it possible to know more about realizations of these arrangements building up the research later on?.

Answer: We change “central planner” to a social planner” and deleted “central hub”. We assume that the decentralized systems can be regulated by social planner(s) (regulatory agency, regional planners, cross-sectoral authorities) interested in the improvement of sectoral/regional performance developing various guidelines for “decentralization” to work. For example, EU Commission develops various regulations for energy, agricultural, water sector activities. The developed linkage procedure can be used by any social planner (decision-makers or regulatory agency (agencies), who attempt to achieve the best result for all sectors/regions (parties) involved. More detailed discussion is beyond this paper.

  1. Lines 145-154. It seems there is repetition of bibliography review and lines 78-84 in the Introduction. Please avoid repeating same topics and delete this second description.

Answer: The repetitions have been deleted.

  1. Lines 260-263. Please explain why the selected subsets add better value to the investigation than other categorizations. How coal can be either primary energy source and also secondary? How is the quota considered in a case like this?. How are solar and wind resources treated?, maybe through power generation capabilities?, please reorganize and explain solid foundations of this initial platform for subsequent research.

Answer: The model is able to include different energy resources depending on a case study. We revised the sentence in the following way: “The structure of the model is such that it can incorporate various energy resources as, e.g., coal, gas, crude oil, renewables. Primary energy sources include coal, crude oil, gas, solar, wind, etc.; secondary energy sources are fuel oil, methanol, hydrogen, electricity, ammonia, etc.; final energy products are coal, fuel oil, gas, hydrogen, ammonia, methanol, electricity, etc.”

Depending on a case study, the subsets can vary. The wind and solar resources require specific sub-models, which are not discussed in this publication. We excluded the “wind and solar” from the sentence to avoid additional explanations.

  1. Lines 285-287. Please add supporting evidence to the expression: … regional planners (…..) pursue a goal to minimize costs and maximize profits……Why should they do that. Please explain or delete.

Answer: We revised the sentence. Now we say, we “assume, social planners (…..) pursue a goal to minimize costs and maximize profits”.  

  1. It is quoted along the document that ‘net profit’ is one main benchmarks for reference, however, there is no explanation or definition of this concept. This is again a very serious barrier to understand the document.

Answer: We revised the sentence and included the definition of net profits as follows: “In this model, the net profits are defined as the amount of money left after subtracting production costs from total profit. In a more general case, the net profits can account for taxes, interest, and other expenses.”

Reviewer 3 Report

This paper has considered integrated LP problems with asymmetric information where each sub-LP problem cannot share its full information, such as objectives and constraints, and further developed a linking-based distributed algorithm to address the incomplete information challenges. Convergence of the algorithm is proved and numerical experiments are provided to verify the convergence. However, the following issues should be addressed before this paper can be accepted. 

  1. No discussions on optimal transport.
    1. Optimal transport is a well-studied area on resource allocation and matching, and there is no discussion on how this paper is related to optimal transport. 
      1. Villani, Cédric. Optimal transport: old and new. Vol. 338. Berlin: Springer, 2009.
      2. Galichon, Alfred. Optimal transport methods in economics. Princeton University Press, 2016.
  2. Missing references on recently developed distributed algorithms for LP, I would suggest the authors add discussions on distributed linear programming and comment on how their approach is different from the existing approaches.
    1. Hughes, Jason, and Juntao Chen. "Fair and distributed dynamic optimal transport for resource allocation over networks." In 2021 55th Annual Conference on Information Sciences and Systems (CISS), pp. 1-6. IEEE, 2021.
    2. Richert, Dean, and Jorge Cortés. "Robust distributed linear programming." IEEE Transactions on Automatic Control 60, no. 10 (2015): 2567-2582.
    3. Ranganathan, Prakash, and Kendall E. Nygard. Distributed Linear Programming Models in a Smart Grid. Springer International Publishing, 2017.

Author Response

Manuscript: sustainability-1524279

Title: Linking distributed optimization models for food, water, energy security nexus management

Authors: Yuri Ermoliev, Anatolij G. Zagorodny, Vjacheslav L. Bogdanov, Tatiana Ermolieva, Petr Havlik, Elena Rovenskaya, Nadejda Komendantova, Michael Obersteiner

Answers to Reviewer.

Thank you for the valuable comments and suggestions, which led to the improvement of our paper. Below please find detailed point-by-point explanations on how we addressed these comments and revised the paper.

Point by point answers to the reviewer:

Comments and Suggestions for Authors

This paper has considered integrated LP problems with asymmetric information where each sub-LP problem cannot share its full information, such as objectives and constraints, and further developed a linking-based distributed algorithm to address the incomplete information challenges. Convergence of the algorithm is proved and numerical experiments are provided to verify the convergence. However, the following issues should be addressed before this paper can be accepted. 

  1. No discussions on optimal transport.
    1. Optimal transport is a well-studied area on resource allocation and matching, and there is no discussion on how this paper is related to optimal transport. 
      1. Villani, Cédric. Optimal transport: old and new. Vol. 338. Berlin: Springer, 2009.
      2. Galichon, Alfred. Optimal transport methods in economics. Princeton University Press, 2016.

Answer: Thank you for this important comment. Indeed, optimal transport methods and transportation theory are important areas in production and resource allocation. We included the suggested references in the text and in the list of references. However, our approach to sectoral/regional models’ linkage under joint constraint (e.g., on common resource or total production) is based on an iterative “decentralized” optimization procedure when sectors/regions optimize independently and in parallel their goal functions under sectoral constraints and quotas. If joint constraints are not fulfilled, the social planner suggests to the systems to revise their quotas (resource use, production, etc.) according to shadow prices of respective constraints, to fulfill the joint constraints. The subsystems, as well as the social planner, do not have full information on models’ specification.

Therefore, the overall goal function of the “linkage” problem cannot be presented as a sum of individual goal functions as it is often done in optimal transport methods. The linkage procedure for “decentralized” regulation of systems is formulated as a nonsmooth optimization model, which is solved following a simple iterative subgradient algorithm. More detailed comparison between the described linkage procedure and transport methods is beyond the scope of this paper.

  1. Missing references on recently developed distributed algorithms for LP, I would suggest the authors add discussions on distributed linear programming and comment on how their approach is different from the existing approaches.
    1. Hughes, Jason, and Juntao Chen. "Fair and distributed dynamic optimal transport for resource allocation over networks." In 2021 55th Annual Conference on Information Sciences and Systems (CISS), pp. 1-6. IEEE, 2021.
    2. Richert, Dean, and Jorge Cortés. "Robust distributed linear programming." IEEE Transactions on Automatic Control 60, no. 10 (2015): 2567-2582.
    3. Ranganathan, Prakash, and Kendall E. Nygard. Distributed Linear Programming Models in a Smart Grid. Springer International Publishing, 2017.

Answer: We included the references and a short discussion of the suggested reference.

Round 2

Reviewer 2 Report

Dear authors, thank you very much for the efforts made. The result is now much more fit for purpose, I believe.

I can accept all your answers, with only one main observation: question 2, seems still to be not clearly answered within the manuscript: could you please make sure that you include in the Abstract and in the Introduction Section the main message in your answer to question 2? i.e.This
paper develops a procedure to link and optimize distributed sectoral and/or regional optimization models thus providing a means of decentralized cross-sectoral coordination in the situation when “private” information about sectoral/regional models’ is not available or it cannot be shared by
modeling teams (sectoral agencies). Thus, the linkage methodology enables to investigate policies in interdependent systems in a “decentralized” fashion. This will enable more stable and  resilient systems’ performance and resource allocation as compared to the independent policies designed by separate models without accounting for interdependencies.   
The paper illustrates the application of the methodology to link detailed energy and agricultural production planning models under joint constraints on water and land use. In addition, the joint constraints can impose restriction on total energy production by energy sector (electricity, gas,
diesel, etc.) and land use sector (biodiesel, methanol); total energy use by energy and agricultural sectors; total agricultural production by distributed farmers/regions, etc. Therefore, the discuss about “energy and natural resources dynamics in general” is beyond the scope of this paper and
can be a topic of another paper.    

Thanks,

Author Response

Manuscript: sustainability-1524279

Title: Linking distributed optimization models for food, water, energy security nexus management

Authors: Yuri Ermoliev, Anatolij G. Zagorodny, Vjacheslav L. Bogdanov, Tatiana Ermolieva, Petr Havlik, Elena Rovenskaya, Nadejda Komendantova, Michael Obersteiner

Reviewer’s comments:

Dear authors, thank you very much for the efforts made. The result is now much more fit for purpose, I believe.

I can accept all your answers, with only one main observation: question 2, seems still to be not clearly answered within the manuscript: could you please make sure that you include in the Abstract and in the Introduction Section the main message in your answer to question 2? i.e.

This paper develops a new approach to link and optimize distributed sectoral and/or regional optimization models thus providing a means of decentralized cross-sectoral coordination in the situation when “private” information about sectoral/regional models’ is not available or it cannot be shared by
modeling teams (sectoral agencies). This will enable more stable and  resilient systems’ performance and resource allocation as compared to the independent policies designed by separate models without accounting for interdependencies.   
The paper illustrates the application of the methodology to link detailed energy and agricultural production planning models under joint constraints on water and land use. In addition, the joint constraints can impose restriction on total energy production by energy sector (electricity, gas,
diesel, etc.) and land use sector (biodiesel, methanol); total energy use by energy and agricultural sectors; total agricultural production by distributed farmers/regions, etc. Therefore, the discuss about “energy and natural resources dynamics in general” is beyond the scope of this paper and can be a topic of another paper.    

Answers to Reviewer.

Dear Reviewer,

Thank you very much for your detailed comments and suggestions, which allowed us to improve our paper and also provided us with additional potential ideas and topics for future research and papers.

We revised the abstract according to your suggestion as follows:

Abstract: Traditional integrated modeling (IM) is based on developing and aggregating all relevant (sub)models and data into a single integrated linear programming (LP) model. Unfortunately, this approach is not applicable for IM under asymmetric information (ASI), i.e., when “private” information about sectoral/regional models is not available or it cannot be shared by modeling teams (sectoral agencies). The lack of common information about LP submodels makes LP methods inapplicable for integrated LP modeling. The aim of this paper is to develop a new approach to link and optimize distributed sectoral/regional optimization models providing a means of decentralized cross-sectoral coordination in the situation of ASI. Thus, the linkage methodology enables to investigate policies in interdependent systems in a “decentralized” fashion. For the linkage, the sectoral/regional models don’t need recoding or reprogramming. They also don’t require additional data harmonization tasks. Instead, they solve their LP submodels independently and in parallel by a specific iterative subgradient algorithm for nonsmooth optimization. The submodels continue to be the same separate LP models. A social planner (regulatory agency) only needs to adjust the joint resource constraints to simple subgradient changes calculated by the algorithm. The approach enables more stable and resilient systems’ performance and resource allocation as compared to the independent policies designed by separate models without accounting for interdependencies. The paper illustrates the application of the methodology to link detailed energy and agricultural production planning models under joint constraints on water and land use.

Also, we included your suggestions and revised as follows the paragraph:

“In this paper we consider the problem of linking sectoral and/or regional linear programming (LP) models into a cross-sectoral integrated model (IM) in the presence of joint constraints when “private” information about sectoral/regional models’ is not available or it cannot be shared by modeling teams (sectoral agencies), i.e., under asymmetric information (ASI). Such linkage is necessary for producing truly integrative management scenarios especially when sectors utilize and compete for common resources, act under joint regulatory constraints or environmental mandates. The approach provides a means of decentralized cross-sectoral coordination and enables to investigate policies in interdependent systems in a “decentralized” fashion. This enables more stable and resilient systems’ performance and resource allocation as compared to the independent policies designed by separate models without accounting for interdependencies.”

We revised the last paragraph of the introduction:

The paper is organized as follows. Section 2 discusses the problem of models’ linkage under joint constraints. Section 2.1 presents a short overview and main shortfalls of several existing approaches, section 2.2. formulates the problem of distributed LP models’ linkage in the presence of joint resource constraints and ASI, section 2.3 outlines the linkage solution procedure based on the parallel solving of equivalent nonsmooth optimization model following a simple iterative subgradient algorithm. Details and main properties of the algorithms are presented in Annex. Section 3 illustrates the application of the methodology to link detailed energy and agricultural production planning models under joint constraints on water and land use. In addition, the joint constraints can impose restriction on total energy production by energy sector (electricity, gas, diesel, etc.) and land use sector (biodiesel, methanol); total energy use by energy and agricultural sectors; total agricultural production by distributed farmers/regions, etc. Section 4 concludes and outlines potential further extensions of the approach, for example, to include more details of energy and natural resources dynamics in general.

In Conclusions, we revised as follows:

… In the paper we referred to linking regional and/or sectoral models. More generally, the problem can address the linkage of models at different spatial and temporal resolutions (e.g., local-global, considering more details of energy and natural resources dynamics in general). Therefore, the linkage problem can be formulated much more generally in terms of sub-models and integrated models, and the approach presented in this paper can still be applicable.

Reviewer 3 Report

My previous comments have been fully addressed. 

Author Response

Manuscript: sustainability-1524279

Title: Linking distributed optimization models for food, water, energy security nexus management

Authors: Yuri Ermoliev, Anatolij G. Zagorodny, Vjacheslav L. Bogdanov, Tatiana Ermolieva, Petr Havlik, Elena Rovenskaya, Nadejda Komendantova, Michael Obersteiner

Reviewer’s comments:

Comments and Suggestions for Authors: My previous comments have been fully addressed. 

Answers to Reviewer.

Dear Reviewer,

Thank you very much for your comments and suggestions, which allowed us to improve our paper and also provided us with additional potential topics for future research and papers.
